# PP2A Functions during Mitosis and Cytokinesis in Yeasts

**DOI:** 10.3390/ijms21010264

**Published:** 2019-12-30

**Authors:** Yolanda Moyano-Rodriguez, Ethel Queralt

**Affiliations:** Cell Cycle Group, Institut d’Investigació Biomèdica de Bellvitge-IDIBELL, Av. Gran Via de L’Hospitalet 199-203, L’Hospitalet de Llobregat, E-08908 Barcelona, Spain; ymoyano@idibell.cat

**Keywords:** yeast, phosphatase, PP2A, cell cycle, mitosis, cytokinesis

## Abstract

Protein phosphorylation is a common mechanism for the regulation of cell cycle progression. The opposing functions of cell cycle kinases and phosphatases are crucial for accurate chromosome segregation and exit from mitosis. Protein phosphatases 2A are heterotrimeric complexes that play essential roles in cell growth, proliferation, and regulation of the cell cycle. Here, we review the function of the protein phosphatase 2A family as the counteracting force for the mitotic kinases. We focus on recent findings in the regulation of mitotic exit and cytokinesis by PP2A phosphatases in *S. cerevisiae* and other fungal species.

## 1. Introduction

Protein phosphatases type 2A (PP2A) are a group of abundant protein phosphatases present in all organisms with conserved structure among eukaryotes. PP2A phosphatases are involved in a myriad of essential processes, including cell growth, proliferation, and cell cycle regulation. PP2A forms a heterotrimeric complex composed of a C catalytic subunit, a scaffold subunit (A), and a regulatory subunit (B) (reviewed in [1,2]). In yeast, the PP2A phosphatase is composed by the scaffold protein Tpd3 (in *S. cerevisiae*) and Paa1 (in *S. pombe*) [3], one of the related catalytic subunits Pph21-22 (in *S. cerevisiae*) and Ppa1-2 (in *S. pombe*), and one of the three regulatory subunits: a 55 KDa regulatory B subunit Cdc55 in *S. cerevisiae* and Pab1 in *S. pombe*, the 56 KDa regulatory B’ subunit Rts1 in *S. cerevisiae* and Par1 and Par2 in *S. pombe*, or the predicted B-subunit Rts3 in *S. cerevisiae*. Cdc55 was first described as the regulatory subunit of the PP2A phosphatase in mammalian cells, where B55α subunit has a 53% of homology with Cdc55 [4,5]. Rts1 was described to be the homolog of B56 in humans [6]. The regulatory subunit confers specificity to the substrates and determines the subcellular localization of the PP2A phosphatase [4,7].

In *Saccharomyces cerevisiae*, Cdc55 and Tpd3 are found at the same subcellular localizations at the cytoplasm and the nucleus throughout the cell cycle (Figure 1). During G1/S, they are also detected at the cortex of the new bud upon bud emergence. In cytokinesis, they are mobilized to the division site. At the bud neck, their localization was seen to be dynamic: first at the daughter side, followed by two rings that converge in one after septation [8]. Conversely, Rts1 is localized at the cytoplasm, the nucleus, and at the spindle pole bodies. The differential localization of the PP2A regulatory subunits controls substrates accessibility and, therefore, confers a specific function to the holoenzyme.

In the case of *Schizosaccharomyces pombe*, the B regulatory subunit, Pab1, is localized at the cytoplasm, the nucleus, and the mitotic spindle [9]. The two B’ regulatory subunits, Par1 and Par2, have a differential subcellular localization. During most of the cell cycle, Par1 is localized at the cytoplasm and the nucleus and translocate to the division site in late mitosis [10]. Par2 protein is first located in one tip of the cell in G1, and, later on, it is localized at the two tips during axial growth. At the end of the cell cycle, during late anaphase and cytokinesis Par2, it is also localized at the division site [11].

The assembly of the PP2A subunits to form the heterodimer or heterotrimer is regulated by post-translational modifications. The methylation/demethylation events of the catalytic subunit Pph1-2 regulating the formation of the heterodimer between the A and C subunits [12] are the best studied. Carboxyl methylation of the C subunit by Ppm1 in budding yeast or Pmt1 in mammals is required in order to interact with the scaffold A subunit and be functional as a holoenzyme [13]. The methylesterase Pme1 counteracts the C subunit methylation and protects PP2A from its degradation [14,15]. On the other hand, the phosphorylation of the Pph21 catalytic subunit at the residues threonine364 or tyrosine367 of the conserved C-terminal sequence TPDYFL is necessary for the assembly to the Cdc55 regulatory subunit [13]. The phosphorylation of PP2A regulatory subunits also regulate PP2A activity. For instance, Rts1 B’ regulatory subunit is phosphorylated in vivo [6], and Cdc55 B subunit has been described to be inhibited by Cdk1-dependent phosphorylation during anaphase [16] (next section).

PP2A are involved in different processes during the cell cycle, including nutrient response, polarized growth, and cell division [17]. Furthermore, PP2A are important regulators of mitosis in most eukaryotes. In budding yeast, the Cdc14 phosphatase has been considered a major source of protein dephosphorylation during mitotic exit [18,19]. However, in other organisms, the PP2A has a prominent role during mitosis [20,21,22,23]. Recently, it has been described that Cdc14 and PP2A cooperate acting in part redundantly, but also in specific and additive ways contributing to the phosphorylation changes that orchestrate mitotic exit [24]. Hence, not only Cdc14 but also other phosphatases, like PP2A, are important for mitosis progression. 

PP2A^Cdc55^ and PP2A^Rts1^ have different roles during mitosis and cytokinesis. PP2A^Cdc55^ regulates Cdc14 activation during mitosis by counteracting phosphorylations of the Cdc14 inhibitor Net1 [25] and the anaphase-promoting complex (APC) subunits [26], while PP2A^Rts1^ controls chromosome biorientation [27,28] and regulates septin reorganization during cytokinesis [29]. PP2A^Rts1^ and, in less proportion, PP2A^Cdc55^ play a crucial role in chromosome bipolar attachment that is discussed in another chapter of this Special Issue Protein Phosphatases and Cell Cycle Regulation in Yeasts [30]. Here, we will focus in the mitotic exit and cytokinesis roles of the PP2A heterotrimers.

## 2. Mitosis Exit Regulation by PP2A

The activation of the APC/C^Cdc20^ cyclosome is required for the initiation of mitotic exit by targeting several proteins for degradation [31]. At the mitotic exit, APC/C^Cdc20^ promotes proteasomal destruction of cyclin B inactivates mitotic Cdk1 [32]. APC/C^Cdc20^ activation is also required for the ubiquitin-dependent degradation of securin [33,34,35] (Pds1 in budding yeast), the separase (Esp1) inhibitor, promoting the metaphase to anaphase transition. Active separase promotes sister chromatids segregation by cleaving the Scc1 subunit of the cohesin complex and triggers mitotic exit through Cdc14 activation [36,37]. PP2A^Cdc55^ prevents the untimely activation of the mitotic exit in different ways: by the adaptation to the spindle assembly checkpoint regulating the cohesin cleavage and by inhibiting Cdc14 release from the nucleolus (Figure 2).

### 2.1. The APC Dephosphorylation by PP2A^Cdc55^

The availability of the co-factor Cdc20 to the APC/C determines the metaphase-to-anaphase transition, and it is regulated by the spindle assembly checkpoint (SAC). The SAC ensures that the mitotic spindles are attached properly to the kinetochores, the chromosomes are correctly aligned to the metaphase plate, and the tension due to the bipolar attachment of the sister chromatids at the metaphase plate is produced [38,39]. The mitotic checkpoint complex (MCC) proteins determine the availability of Cdc20 to the APC. Moreover, APC/C^Cdc20^ activation is also regulated by phosphorylation. Cdc28–Clb2 phosphorylates the APC subunits Cdc16, Cdc23, and Cdc27 upon spindle damage conditions to activate APC [40,41]. In fact, it was described that the phospho-null mutants for these proteins [42] and the inactivation of Cdc28 [43] impaired APC/C^Cdc20^ activity. Conversely, PP2A^Cdc55^ counteracts the Cdk1 phosphorylation of the APC/C subunit Cdc16 [26,43], keeping SAC active until the cell is prepared for anaphase. Tight balance between Cdc28–Clbs and PP2A^Cdc55^ activities is important for the adaptation to the spindle checkpoint [43]. Consistently, it was observed that in the absence of Cdc55, the cells bypass the SAC and become insensitive to the microtubule instability induced by nocodazole addition [41,44,45]. Cdc55 is also required for proper cell cycle delay in response to tensionless attachment [46] suggesting a role in chromosome bipolar attachment. PP2A^Cdc55^ contributes indirectly to the prevention of untimely mitotic entry by inhibiting premature Cdc14 release from the nucleolus and precocious cleavage of sister chromatid cohesin (discussed below).

### 2.2. The Regulation of the Cohesin Cleavage by PP2A^Cdc55^

PP2A^Cdc55^ also regulates anaphase onset by counteracting the phosphorylation of the Scc1 subunit of the cohesin complex [47]. Scc1 is phosphorylated by the polo-like kinase Cdc5, promoting the Scc1 cleavage by separase [48,49,50,51]. Before anaphase, the dephosphorylation of the Scc1 by PP2A^Cdc55^ prevents its recognition by separase, avoiding premature sister chromatids segregation [47]. Premature cohesin cleavage might be responsible for the impaired chromosome bipolar attachment observed in absence of Cdc55 [52]. In early anaphase, upon separase downregulation of PP2A^Cdc55^, Scc1 dephosphorylation is inhibited, promoting cohesin cleavage. Separase regulates Scc1 directly by cleaving it, and also indirectly, through the regulation of Scc1 dephosphorylation by PP2A^Cdc55^ inhibition.

### 2.3. The FEAR-Cdc14 Release by PP2A^Cdc55^

The first described mitotic function of PP2A^Cdc55^ was its role in the activation of the phosphatase Cdc14 during anaphase. Cdc14 was proposed as the major Cdk-counteracting phosphatase in budding yeast mitotic exit [18,19]. Cdc14 is kept sequestered in the nucleolus by its binding to the nucleolar protein Net1 during most of the cell cycle. The Cdc14 release from the nucleolus during anaphase is required for its activation. At anaphase, Net1 is phosphorylated by Cdk1–Clb2 and Cdc5 mitotic kinases [53,54,55]. Phosphorylated Net1 has low affinity toward Cdc14, and the phosphatase is translocated, first to the nucleus during early anaphase, and then to the cytoplasm in late anaphase. Early anaphase Cdc14 release is regulated by separase in conjunction with a series of proteins (Slk19, Spo12, Fob1, Cdc5, Cdk1–Clb2, Cdc55, and Hit1) [25,53,56,57,58,59,60,61] commonly known as the FEAR pathway (the cdcfourteen early anaphase release (FEAR)).

Clb2–Cdc28 complex has a peak of activity at metaphase when it phosphorylates many mitotic proteins, including Net1 [53]. During most of the cell cycle, Net1 phosphorylation is counteracted by PP2A^Cdc55^ [25,41], and, as a consequence, Cdc14 is sequestered at the nucleolus. In *cdc55Δ* mutant cells, Net1 is phosphorylated already at metaphase, and Cdc14 is prematurely released from the nucleolus. By contrast, in *rts1Δ* cells Cdc14 was released with similar kinetics to the wild-type cells [25]. In addition, it was shown that PP2A^Cdc55^ has phosphatase activity against Net1 in vitro [16,25] and both proteins co-immunoprecipitate in vivo [16], suggesting that Net1 is a substrate of PP2A^Cdc55^. During early anaphase, downregulation of the PP2A^Cdc55^ phosphatase activity allows the accumulation of the Cdk1–Clb2-dependent Net1 phosphorylation and promotes the Cdc14 release from the nucleolus [25]. Remarkably, the anaphase-specific inhibition of the PP2A^Cdc55^ phosphatase activity is due to phosphorylation of the regulatory subunit Cdc55 by Cdk1–Clb2 and depends on active separase and Zds1 proteins [16,25,56,58]. Separase is a cysteine-like caspase protease with proteolytic and non-proteolytic functions during mitosis [36,37]. Separase regulates sister chromatids segregation by cleaving the Scc1 subunit of the cohesin complex and triggers mitotic exit through Cdc14 activation. At anaphase onset, Zds1/2 proteins and separase cooperatively trigger mitosis exit by the downregulation of PP2A^Cdc55^ [58]. The Zds1 and Cdc55 interaction is mediated by the Zds1 C-terminal Zds_C motif, and it is required for the nucleolar Cdc55 localization [56].

PP2A^Cdc55^ is also required for the proper temporal initiation of meiotic events [62]. Similar to mitosis, PP2A^Cdc55^ also regulates the FEAR pathway during meiosis [63,64]. PP2A^Cdc55^ dephosphorylates Net1 and promotes Cdc14 release from the nucleolus, preventing precocious exit from meiosis I. In addition, PP2A^Cdc55^ is required for reductional chromosome segregation in the absence of recombination independently of its role in the FEAR pathway [65]. 

### 2.4. MEN (SIN) Regulation by PP2A

Cdc14 activation and release during anaphase is mediated by two parallel pathways: the FEAR and the mitotic exit network (MEN) [66]. MEN (also known as the Hippo pathway in higher eukaryotes) is a GTPase-driven signaling cascade associated with the centrosomes (spindle pole body; SPB in yeast) that regulates mitotic exit, enables the control of the spindle orientation, and promotes cytokinesis in budding yeast. 

The core of the MEN cascade consists of two serine/threonine kinases, Cdc15 (PAK kinase in higher eukaryotes) and Dbf2-Mob1 (LATS kinase in higher eukaryotes). They are activated in mid–late anaphase to maintain Cdc14 released from the nucleolus and promote its full activation [67,68,69]. MEN activation depends on the first element of the cascade: the small Ras-like GTPase Tem1. During an unperturbed cell cycle, Bub2/Bfa1, the MEN inhibitor, keeps Tem1 inactive. PP2A^Cdc55^ contributes to keep Bub2/Bfa1 active by dephosphorylating Bfa1 in metaphase [70]. When cells reach anaphase with a correct aligned mitotic spindle, Cdc5 phosphorylates Bfa1 and inactivates the Bub2/Bfa1 GAP activity [71,72]. The anaphase-specific inactivation of PP2A^Cdc55^ also contributes to increase the Cdc5-dependent Bfa1 phosphorylation and promotes the activation of MEN. Therefore, PP2A^Cdc55^ not only facilitates the FEAR-dependent Cdc14 release in early anaphase but also contributes to alleviation of the MEN inhibitory signal imposed by Bfa1/Bub2.

Once Tem1 is active, it interacts with the Pak-like kinase Cdc15 [73], which in turn phosphorylates and activates the kinase subunit, Dbf2, of the LATS-related kinase Dbf2-Mob1. Upon activation, Dbf2-Mob1 promotes the full activation of the Cdc14 phosphatase in mid/late anaphase [67,68,74,75]. In addition, most of the MEN proteins are regulated by phosphorylation, making MEN activity restrained by Cdk1 and stimulated by the action of the opposing phosphatases, Cdc14 and PP2A^Cdc55^ (Figure 3). Cdk1 restrains MEN activity through Cdc15 and Mob1 phosphorylation [76]. At anaphase, Cdc15 is dephosphorylated by the FEAR-released Cdc14 facilitating its activation [77,78,79,80,81]. Mob1 dephosphorylation at late anaphase is necessary for Dbf2-Mob1 activation. Abrupt Cdk1 inactivation and Cdc14 release from the nucleolus contribute to Mob1 dephosphorylation in late anaphase [76]. In addition, PP2A^Cdc55^ also dephosphorylates Mob1 protein [70]. At anaphase onset, PP2A^Cdc55^ downregulation facilitates Cdk1-dependent phosphorylation of Mob1, contributing to Dbf2–Mob1 inhibition. During exit from mitosis, PP2A^Cdc55^ reactivation could promote Mob1 dephosphorylation supporting Dbf2–Mob1 activation. 

Asymmetrically diving cells, like budding yeast, require a tight control of the mitotic spindle alignment along the mother–daughter cell axis and perpendicular to the division plane for accurate chromosome segregation. The spindle position checkpoint (SPOC) delays mitosis progression when the orientation of the mitotic spindle is incorrect. The activation and functionality of SPOC depend on the ability of Bub2/Bfa1 to inhibit MEN. Kin4 kinase phosphorylates and activates Bfa1 when the spindle is misaligned, preventing anaphase progression by the activation of SPOC [82,83,84]. Phosphorylation of Bfa1 by Kin4 prevents the Cdc5-dependent phosphorylation of Bfa1, keeping MEN inactive [72,82,85,86]. Although PP2A^Cdc55^ is the main PP2A regulating mitotic exit, PP2A^Rts1^ was also described to regulate MEN upon activation of the SPOC. PP2A^Rts1^ is a SPOC component acting upstream of Kin4. PP2A^Rts1^ dephosphorylates Kin4, regulating the association of Kin4 to the SPBs, and thereby restraining MEN activity [87]. Upon proper spindle alignment, Kin4 is inactivated by Lte1, and Bfa1 is phosphorylated by Cdc5, promoting MEN activation [86,88].

The MEN pathway is closely related to the septation initiation network (SIN) in *Schizosaccharomyces pombe* and the Hippo pathway in mammals. Their most conserved role is the regulation of cytokinesis (see below). The upstream SIN effector is the small GTPase Spg1 (Tem1) and is controlled by the GTPase-activating protein (GAP) Byr4-Cdc16 (Bub2/Bfa1) and Etd1 the GTP/GDP exchange factor (GEF) (review in [89]). At the core of the SIN pathway, Sid1-Cdc14 is the PAK-like kinase (Cdc15) and Sid2-Mob1 is the LATS-kinase (Dbf2-Mob1). Two additional kinases, the polo-like Plo1 and the Ste20-family Cdc7 are also part of the SIN pathway. Mutations of the PP2A regulatory subunits (Pab1 and Par1) and the major catalytic subunit Ppa2 rescue conditional SIN mutants [9,10,90], suggesting that PP2A inhibits SIN signaling (Figure 2). PP2A^Pab1^ inhibits the SIN pathway in parallel to Etd1 [9], and, similar to *S. cerevisiae*, it has been proposed that the main candidate to be the target of PP2A^Pab1^ is the GAP Byr4-Cdc16 [9]. The other PP2A holoenzyme, PP2A^Par1/2^, regulates the localization of the Cdc7 kinase inhibiting SIN, in order to avoid multiple rounds of septation [10,90,91].

Finally, PP2A^Pab1^ and PP2A^Par1^ are also regulated by the PP1-like phosphatase Dis2. PP1 binds to and activates PP2A^Pab1^ through a conserved RVXF motif present in Pab1, the B55 subunit. Active PP2A^Pab1^ dephosphorylates Par1 and promotes PP1 recruitment, which in turn further activates PP2A^Par1^ phosphatase. In this way, PP1-induced activation of both PP2A^B55^ and PP2A^B56^ coordinates mitotic progression and exit from mitosis [92].

## 3. The Role of PP2A in Cytokinesis

In budding yeast, mitotic Cdk1 inhibits a second round of DNA replication and cytokinesis to ensure that cytokinesis occurs only after chromosome segregation is completed. Therefore, cells need to inactivate Cdk1 activity and dephosphorylate mitotic proteins at the end of mitosis in order to trigger cytokinesis. MEN activation during anaphase leads to the downregulation of Cdk1, exit from mitosis, and onset of cytokinesis through the activation of the Cdc14 phosphatase [18,93,94]. Cytoplasmic Cdc14 dephosphorylates key targets as the APC/C coactivator Cdh1 that promotes degradation of the cyclins B and the Swi5 transcription factor, which induces expression of the Cdk1 inhibitor Sic1 [93]. Moreover, Cdk1 inactivation triggers the accumulation of the MEN proteins Cdc15, Dbf2-Mob1, Cdc5, and Cdc14 at the bud neck [69,94,95,96,97]. The MEN’s role in cytokinesis is the most conserved function of the MEN orthologs. The SIN pathway in *S. pombe* controls septum formation and contractile ring assembly [98,99], and the Hippo pathway regulates actin polymerization during cytokinesis [100]. In budding yeast, MEN-released Cdc14 dephosphorylates cytokinetic proteins such as Iqg1, Inn1, and Chs2 [101,102,103], regulating primary septum formation and ingression of the plasma membrane [101,103,104].

PP2A was suggested to be involved in cytokinesis based on the multinucleated and elongated phenotype of the *rts1Δ* mutant at high temperature [4,6]. Moreover, Cdc55 [8] and Rts1 [29] were described to be present at the division site during cytokinesis. Consistently, PP2A^Rts1^ was involved in the dephosphorylation of the septin Shs1, regulating septin dynamics during cytokinesis [29]. On the contrary, although some cytokinetic proteins have been suggested to be new PP2A^Cdc55^ substrates [105], a direct role during cytokinesis has not been described. Further work is required to clarify the putative PP2A^Cdc55^ role during cytokinesis. 

In addition, PP2A^Rts1^ regulates the cell cycle entry into the next G1 by inhibiting the transcription factor Ace2 [27,106]. Rts1 is required for proper phosphorylation and localization of Ace2. Ace2 regulates the gene expression of the septum hydrolases required for the physical separation of the two new cells during cytokinesis [107,108,109]. Lack of Rts1 provokes higher Ace2 localization at the mother cell nucleus, affecting cytokinesis progression. 

As previously mentioned, PP2A^Par1^ and PP2A^Pab1^ inactivate the SIN pathway which coordinates mitotic exit with cytokinesis. In addition, PP2A^Pab1^ participates in cytokinesis directly through RhoA regulation. RhoA activity is compromised when Pab1 is overexpressed and the glucan synthesis promoted by RhoA is reduced upon Pab1 deletion [110]. Similarly, PP2A^ParA^ negatively regulates SIN in *A. nidulans* during septation [111]. 

Furthermore, PP2A^Cdc55^ and PP2A^Rts1^ regulate cytokinesis in *Candida albicans* [112]. Deletion mutants of *CDC55* and *RTS1* show an increase in chitin staining and higher sensitivity to calcofluor and caspofungin. Calcofluor binds to chitin and caspofungin inhibits glucan synthesis, the two main components of the yeast septa. An increase in Sep7 septin phosphorylation is observed in absence of Cdc55 and Rts1, suggesting that PP2A regulates cytokinesis in *Candida albicans* through the dephosphorylation of Sep7 [112,113]. Dephosphorylation of the core septin AspB (ortholog of Cdc3 septin) by PP2A-ParA (Rts1 in *S. cerevisiae*) is also involved in septation in the fungal pathogen *Aspergillus fumigatus* [114]. Moreover, ParA and PabA (Cdc55 in *S. cerevisiae*) also play a role in septation in *Aspergillus nidulans* [111,115]. In conclusion, the role of PP2A in cytokinesis and septation seems to be conserved in filamentous fungi. 

## 4. Concluding Remarks

Tight control of both kinases and phosphatases is crucial for regulation of the cell cycle progression and chromosome segregation. PP2A phosphatase regulates multiple functions during the cell cycle in yeast through the formation of different heterotrimeric complexes. Substrate specificity and PP2A subcellular localization are conferred by their regulatory subunits. Although some PP2A substrates have been identified and studied during mitosis and cytokinesis, they still represent a small fraction of PP2A targets, limiting our knowledge of the temporal and spatial control of the PP2A roles. Further studies on PP2A substrates may provide new insights into the mechanisms that coordinate completion of chromosome segregation before cytokinesis.

## Figures and Tables

**Figure 1 ijms-21-00264-f001:**
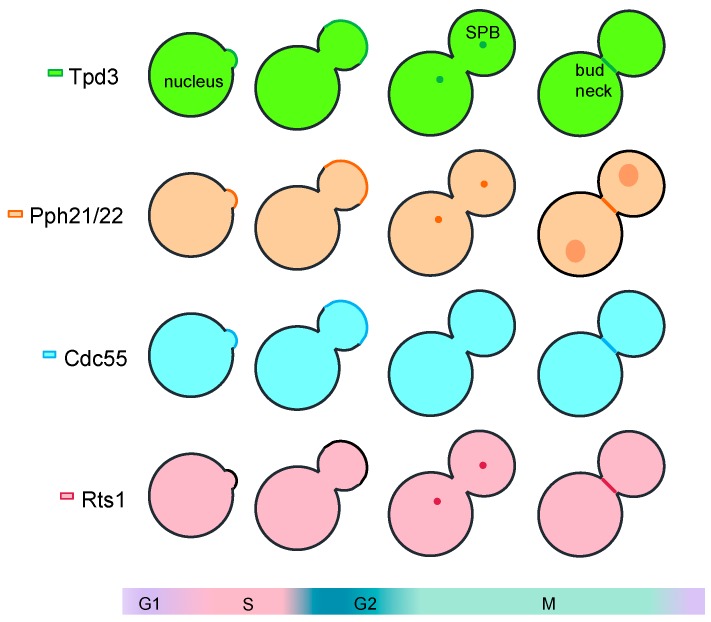
Representation of the subcellular localization of the PP2A-Cdc55 subunits along the cell cycle in *S. cerevisiae*. All the PP2A subunits are found at the nucleus and the cytoplasm throughout the cell cycle and at the division site during cytokinesis. In G1/S, Cdc55, Pph21/2, and Tpd3 are located at the cortex of the new bud. Rts1, Pph21/2, and Tpd3 localized to the SPB during mitosis.

**Figure 2 ijms-21-00264-f002:**
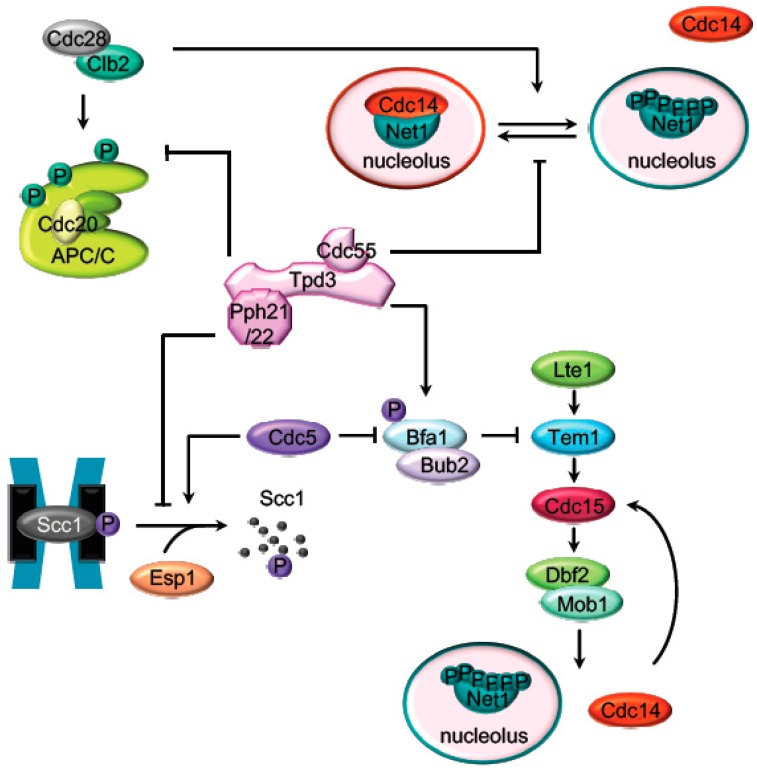
Targets of PP2A^Cdc55^ during mitosis. Representation of the main mitotic PP2A^Cdc55^ substrates described in budding yeast. Before anaphase onset, PP2A^Cdc55^ counteracts the Cdk1 phosphorylation of the APC/C subunits and the Cdc14 inhibitor, Net1. Scc1 dephosphorylation by PP2A^Cdc55^ also prevent premature sister chromatids segregation before anaphase. PP2A^Cdc55^ contributes to keep MEN inactive by counteracting Bfa1 phosphorylation in metaphase.

**Figure 3 ijms-21-00264-f003:**
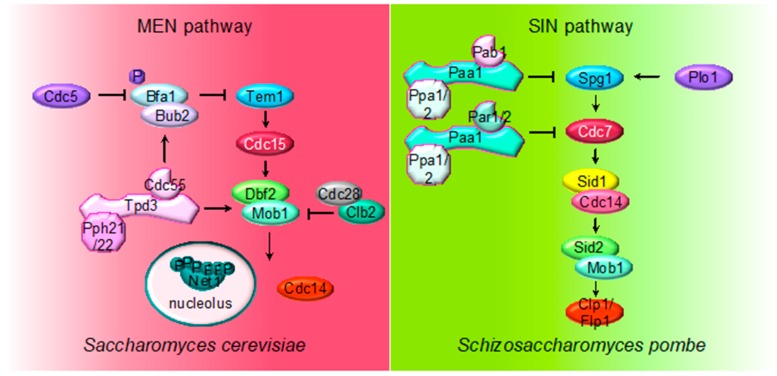
Multiple roles of PP2A regulating MEN and SIN pathways. The cartoon shows the different PP2A holoenzymes and their substrates regulating MEN during mitosis in *S. cerevisiae* and SIN in *S. pombe* during septation. The first element of the cascades is the small Ras-like GTPase Tem1 in *S. cerevisiae* and Spg1 in *S. pombe*. The core of the MEN and SIN cascades consist of two serine/threonine kinases: Cdc15 in *S. cerevisiae* and Sid1-Cdc14 in *S. pombe* (PAK kinase in higher eukaryotes) and Dbf2-Mob1 in *S. cerevisiae* and Sid2-Mob1 in *S. pombe* (LATS kinase in higher eukaryotes). Two additional kinases, the polo-like Plo1 and the Ste20-family Cdc7, are also part of the SIN pathway.

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
