# Peer review of "PP2A Functions during Mitosis and Cytokinesis in Yeasts"

_ijms, 2019, doi:10.3390/ijms21010264_

Round 1

Reviewer 1 Report

This is  a very interesting review of the roles of PP2A in mitosis and cytokinesis. The emphasis is on yeast, mainly budding yeast but also covering fission yeast and Candida.

Here are some suggestions for improvements before publication:

No keywords are provided.

Line 71. The sentence: …At mitotic exit,promotes… could be changed to …At mitotic exit, APC/C-Cdc20 promotes…

Line 74: APC/C-Cdc20 promotes the metaphase to anaphase transition rather than entry into mitosis.

Lines 139-144: Several references are cited by the name of the authors and the year of publication instead of by a number. These references are not present in the list of references.

The list of references should be edited, as many references are incomplete (in many cases the journal chapter and page numbers are missing).

Author Response

We thank the reviewer for the critical reading of the manuscript and the suggestions to improve it.

No keywords are provided. We added the list of keywords

Line 71. The sentence: …At mitotic exit,promotes… could be changed to …At mitotic exit, APC/C-Cdc20 promotes…The sentence have been changed following the reviewer suggestion

Line 74: APC/C-Cdc20 promotes the metaphase to anaphase transition rather than entry into mitosis. The sentence have been changed following the reviewer suggestion

Lines 139-144: Several references are cited by the name of the authors and the year of publication instead of by a number. These references are not present in the list of references. The references have been included and fixed accordingly

The list of references should be edited, as many references are incomplete (in many cases the journal chapter and page numbers are missing). We are sorry about the incomplete sentences. It is the first time that we have problems with Mendeley and we do not understand why Mendeley did not work properly this time. We have fixed the references accordingly.

Reviewer 2 Report

This review covers different aspects of the functions of PP2A in anaphase onset, mitotic exit and cytokinesis. It should be published after some minor issues are addressed.

English editing will be helpful to improve the quality of this review. The title of Figure 1 is “Targets of PP2A-Cdc55 in early anaphase”, but the function of Bfa1-Bub2 is in late anaphase. Thus, the title does not fit well with the content of this figure. Line 87-88: “The SAC components are mainly ……” This sentence is hard to follow. Mps1 is a critical SAC component, but is not a part of MCC. Line 224-245: “PP2A was suggested to …… and cdc55 at low temperature.” Cdc55 mutant is cold sensitive and this phenotype is due to the accumulation of CDK inhibitor Swe1, and deletion of SWE1 suppresses the elongated bud morphology of cdc55 mutant. Thus, the elongated bud morphology may not be an indication of cytokinesis defect in cdc55 mutant.

Minor points:

Line 72: destruction of cyclin B inactivates mitotic CDK.

Line 98: ….contributes indirectly to prevention of untimely…

Line 107: might be responsible for the…

Line 138: Zds1 motif?

Line 160: contributes to alleviation of the

Line 171: PP2A-Cdc55 also dephosphorylates Mob1

Line 173: contributes to Dbe2-Mob1 inhibition.

Line 209: The role of PP2A in cytokinesis

Line246: Caspofungin is a glucan synthesis inhibitor, but it does not bind to glucan.

Line 249: The role of PP2A in cytokinesis….

Author Response

We thank the reviewer for the critical reading of our manuscript and the suggestions to improve it. 

The English have been corrected by and English professional editing service.  The Figure Legend 1 has been fixed following the reviewer instructions. Line 87-88: “The SAC components are mainly ……” This sentence is hard to follow. Mps1 is a critical SAC component, but is not a part of MCC. We deleted the sentence and changed the next sentence accordingly. Line 224-245: “PP2A was suggested to …… and cdc55 at low temperature.” We agree with the reviewer that the elongated phenotype of the cdc55 mutant is not due to cytokinetic defects. The authors in the original paper suggested so, and for this reason we included it. In any case, we have deleted it and fixed the sentence accordingly. Line 138: Zds1 motif? The C-terminal of the Zds1/2 proteins contains a structural motif called Zds_C motif conserved among fungi. We followed the nomenclature of the first time it was described as Zds_C motif. All the other minor points of the reviewer have been also fixed.

Reviewer 3 Report

PP2A is one fundamental enzymatic complex that mediates many processes in eukaryotic cells, most of them related to cell cycle progression and to critical events during cell division. The manuscript submitted by Moyano-Rodriguez and Queralt is a very interesting and updated review summarizing the functions of PP2A complex in yeast during mitosis and cytokinesis. Providing a detailed overview of some very complex but yet poorly understood key mechanisms orchestrated by PP2A and its interaction partners is very challenging and I really liked parts of the review. However, I think the English writing is its weakest point and should be improved throughout. In addition, I have a few comments, which are given below:

-The abstract is very poor and not informative, therefore, it needs to be revised.

-This review describes, almost exclusively, experimental findings concerning PP2A in yeast. PP2A complexes in other eukaryotes perform not only similar, but importantly, many more different functions. Therefore, it has to be clearly mentioned in the title and throughout the text that the review focuses on PP2A functions during mitosis and cytokinesis in fungi/yeast. 

-Along the same lines, the first paragraph of the “concluding remarks” section is a bit irrelevant to the rest manuscript. Moreover, the entire section could be revised/improved. 

-Introduction: the authors might consider adding a schematic illustration of cell cycle stages and the respective localization pattern(s) of PP2A components in yeast.

-Lines 139-144: The style of the references used there is different. Besides, these references are missing from the reference list.

-The authors might consider revising and/or rewriting both figure legends in order to be more comprehensive.

-Lines 242-249: The authors might consider expanding this paragraph and cite some publications describing the interference of PP2A with septation in filamentous fungi species other than Candida.  

Line 9: phosphatases instead of phosphates

Line 26, 27: “While” is not necessary in that sentence.

Line 35: delete the semicolon

Line 38, 39: This sentence might be revised and, once again, the semicolon is not needed.

Line 100: discussed below

Line 106: responsible for instead of: the responsible

Line 147: abbreviation of FEAR has been introduced already in line 121.

Line 189: delete the round bracket  (

Line 230: is required instead of: will be required.

-In my opinion, the frequent use of semicolon in sentences is not necessary.

Author Response

We thank the reviewer for the critical reading of the manuscript and the suggestions to improve it.

The main changes of the revised version of the manuscript:

The English have been edited by a professional English editing service. The Figure legends have been extended and better explained.

The title and the text now include "in yeasts" to clarify that the review focuses on PP2A functions in yeasts. 

A schematic illustration of the localisation pattern of PP2A have been included following the reviewer suggestion.

The problems with the references have been fixed. 

Lines 242-249: The authors might consider expanding this paragraph and cite some publications describing the interference of PP2A with septation in filamentous fungi species other than Candida.  We have added information in other fungi. All the other minor comments have been fixed accordingly